# Incidence and Risk Factors of Cancer in the Anal Transitional Zone and Ileal Pouch following Surgery for Ulcerative Colitis and Familial Adenomatous Polyposis

**DOI:** 10.3390/cancers14030530

**Published:** 2022-01-21

**Authors:** Guillaume Le Cosquer, Etienne Buscail, Cyrielle Gilletta, Céline Deraison, Jean-Pierre Duffas, Barbara Bournet, Géraud Tuyeras, Nathalie Vergnolle, Louis Buscail

**Affiliations:** 1Department of Gastroenterology and Pancreatology, CHU Toulouse-Rangueil (University Hospital Centre) and Toulouse University, UPS, 31059 Toulouse, France; lecosquer.g@chu-toulouse.fr (G.L.C.); gilletta.c@chu-toulouse.fr (C.G.); bournet.b@chu-toulouse.fr (B.B.); 2Department of Surgery, CHU Toulouse-Rangueil and Toulouse University, UPS, 31059 Toulouse, France; ebuscail@me.com (E.B.); duffas.jp@chu-toulouse.fr (J.-P.D.); tuyeras.g@chu-toulouse.fr (G.T.); 3IRSD, Toulouse University, INSERM 1022, INRAe, ENVT, UPS, 31300 Toulouse, France; celine.deraison@inserm.fr (C.D.); nathalie.vergnolle@inserm.fr (N.V.); 4Centre for Clinical Investigation in Biotherapy, CHU Toulouse-Rangueil and INSERM U1436, 31059 Toulouse, France

**Keywords:** anal transitional zone cancer, ileal pouch, ulcerative colitis, familial adenomatous polyposis, high grade dysplasia

## Abstract

**Simple Summary:**

Proctocolectomy with ileal pouch-anal anastomosis is the intervention of choice for ulcerative colitis and familial adenomatous polyposis requiring surgery. However high-grade dysplasia and cancer in the anal transitional zone and ileal pouch after 20 years is estimated to be 2 to 4.5% and 3 to 10% in ulcerative colitis and familial polyposis, respectively. The risk factors for ulcerative colitis are the presence of pre-operative dysplasia or cancer, disease duration > 10 years and severe villous atrophy. For familial polyposis, the risk factors are the number of pre-operative polyps > 1000, surgery with stapled anastomosis and the duration of follow-up. Even if anal transitional zone and ileal pouch cancers seldom occur following proctectomy for ulcerative colitis and familial adenomatous polyposis, the high mortality rate associated with this complication warrants close endoscopic monitoring, mainly every year with pouchoscopy including chromoendoscopy.

**Abstract:**

Proctocolectomy with ileal pouch-anal anastomosis is the intervention of choice for ulcerative colitis and familial adenomatous polyposis requiring surgery. One of the long-term complications is pouch cancer, having a poor prognosis. The risk of high-grade dysplasia and cancer in the anal transitional zone and ileal pouch after 20 years is estimated to be 2 to 4.5% and 3 to 10% in ulcerative colitis and familial polyposis, respectively. The risk factors for ulcerative colitis are the presence of pre-operative dysplasia or cancer, disease duration > 10 years and severe villous atrophy. For familial polyposis, the risk factors are the number of pre-operative polyps > 1000, surgery with stapled anastomosis and the duration of follow-up. In the case of ulcerative colitis, a pouchoscopy should be performed annually if one of the following is present: dysplasia and cancer at surgery, primary sclerosing cholangitis, villous atrophy and active pouchitis (every 5 years without any of these factors). In the case of familial polyposis, endoscopy is recommended every year including chromoendoscopy. Even if anal transitional zone and ileal pouch cancers seldom occur following proctectomy for ulcerative colitis and familial adenomatous polyposis, the high mortality rate associated with this complication warrants endoscopic monitoring.

## 1. Introduction

Restorative proctocolectomy (RPC) with ileal pouch-anal anastomosis (IPAA) is the intervention of choice for familial adenomatous polyposis (FAP) and ulcerative colitis (UC) requiring surgery [1]. UC is characterised by chronic mucosal inflammation of the rectum and colon [2]. The two peaks for UC diagnosis are between 15 and 30 years of age with a second smaller peak after 60 years of age [3]. The colectomy rate 10 years post-diagnosis has recently been estimated to be 15.6% [4]. The main indications for surgery are endoscopically unresectable dysplasia (or cancer), chronic colitis refractory to medical management, complicated (uncontrolled haemorrhage, perforation) and refractory acute severe colitis [5]. FAP is an inherited autosomal dominant disease associated with predisposition to colorectal cancer, with a prevalence of 1 in 10,000 inhabitants [6]. This syndrome is caused by a germline mutation in the adenomatous polyposis coli gene (located on chromosome 5q21) [7]. The phenotypic presentation is characterised by the early onset of hundreds of adenomas leading to colorectal cancer by the age of 40 in almost 100% of cases. Prophylactic colectomy is the only treatment to reduce the colorectal cancer risk and should be offered to all patients between 15 and 25 years of age [8].

RPC consists of the removal of the rectum and the colon and the construction of an ileal pouch (reservoir formed with the last 40 cm of the ileum) followed by IPAA [9]. This surgery can be performed in one, two or three stages depending on the colectomy indication. In patients at high risk of complication (e.g., acute severe colitis), the first step is to remove the right, transverse and left colon (i.e., subtotal colectomy) with double-end ileostomy and sigmoidostomy [10]. Then, 3 to 6 months later, the rectum and sigmoid are removed and an IPAA is formed with a temporary diverting ileostomy to allow the anastomosis to heal. Finally, the stoma is reversed 6 to 8 weeks later. In the absence of local inflammation or factors indicative of complications, a two-stage procedure can be carried out. The first step is RPC with IPAA, and a temporary diverting ileostomy followed 6 to 8 weeks later by closure of the stoma. A single-stage technique is chosen by some surgeons for certain patients (young healthy patients, elective surgery) to avoid ileostomy.

The main complications reported can be divided into short-term (within 1 month after surgery) and long-term complications [11]. Short-term complications are acute bleeding, leakage, abscesses and small bowel obstruction. The long-term complications are incontinence, chronic pelvic sepsis, pouch stricture, decreased fertility, pouch failure (defined as the need to remove the pouch), pouch dysplasia and cancer [12]. Patients operated on for UC are also at risk of pouchitis (i.e., inflammation of the pouch) and development of Crohn’s disease in the pouch. The laparoscopic approach is to be preferred where possible, as it has been proven to lower the risk of postoperative complications and to preserve fertility [13,14].

Ileoanal anastomosis is the critical part of the surgery. Two anastomotic techniques have been described, namely double-stapled or hand-sewn anastomosis (Figure 1) [15]. Manual (hand-sewn) anastomosis consists of creating an anastomosis between the pouch and the anal transitional zone (ATZ) just above the dentate line following endoanal mucosectomy [16,17]. The ATZ is defined as the area between the dentate line and the columnar epithelium, measuring 5–10 mm [18,19]. Histologically, this zone is characterised by the interchange between the squamous epithelium (from the anus) and the columnar epithelium (from the rectum) [20]. Conversely, a mechanical (double-stapled) anastomosis is created 1–2 cm above the ATZ, leaving a cuff of rectal mucosa [21]. The advantage of the manual technique is that any potential risk of diseased tissue (inflammation, dysplasia, cancer) is eradicated. The downside of the technique is an increased risk of leakage [22]. On the other hand, leaving the rectal cuff allows air-liquid/solid differentiation, thus lowering the risk of incontinence, which is a major issue for patients [23,24].

The epidemiology of these diseases implies that some patients undergo surgery at a young age. Balancing functional results and the risk of cancer in the ATZ and ileal pouch after surgery poses a real management challenge. Indeed, the rate of dysplasia of the ATZ after RPC for UC was estimated as 1.13% in a recent meta-analysis [25]. Tsunoda et al. described a higher rate of dysplasia in the case of FAP [26]. Cases of adenocarcinoma of the ATZ and pouch have been reported for both diseases [27,28]. Derikx et al. reported a cumulative incidence of 2% and 6% for dysplasia and cancer of the pouch per se, 10 and 20 years after RPC, respectively, [29]. The prognosis for pouch adenocarcinoma is poor. Wu et al. reported a global mortality rate of 42.9% 2 years after the diagnosis of pouch cancer [30]. Despite this known risk, few follow-up data are available for anastomosis. Those patients must be offered endoscopic follow-up to screen for dysplasia and cancer. The treatment proposed for cancer of the ATZ (or pouch) is pouch excision and terminal ileostomy. Endoscopic resection, notably by submucosal dissection, has emerged as an alternative curative treatment for endoscopically resectable lesions (dysplasia and cancer) [31,32].

The aim of this review is to report on the available data for dysplasia and cancer of the ATZ and pouch focusing on incidence, risk factors, the impact of surgery, in particular, and the initial pathology. This review will also outline the mechanisms leading to dysplasia, including the role of inflammation in the process of carcinogenesis and interactions between mucosa. Finally, we propose an algorithm for post-IPAA endoscopic monitoring.

## 2. Methods

A PubMed literature search for “pouch dysplasia”,” pouch cancer”, “pouch neoplasia”, “anal transitional zone dysplasia”, “anal transitional zone cancer”, “anal transitional zone neoplasia”, “ulcerative colitis”, “familial adenomatous polyposis”, “restorative proctocolectomy,” “ileoanal anastomosis” and “ileal pouches” was conducted. Two authors (GLC and EB) conducted two independent literature reviews, both using the same strategy. All articles from 1978 to December 2021 were included if they reported findings relating to dysplasia or cancer of the pouch and/or the ATZ and/or the rectal cuff (epidemiology, mechanisms, treatments, screening programme). Particular attention was paid to publications including endoscopic follow-up from which valuable conclusions could be drawn.

Additional articles were identified by cross-referencing papers from the initial search. The search excluded articles that were not in English and non-human studies, as well as editorials. Studies on patients suffering from Crohn’s diseases or indeterminate colitis have not been included in this review, given the rarity of the situation and the specific issues [33].

## 3. Dysplasia and Cancer of the Ileal Pouch in Patients with Ulcerative Colitis

### 3.1. Epidemiology and Risk Factors

According to the largest prospective study published, the incidence of ATZ dysplasia was estimated at 4.5% 10 years after surgery [34]. Several cases of adenocarcinoma of the rectal cuff and anastomosis have been reported since the 1990s [35,36,37,38,39,40,41,42]. However, the first case of ATZ adenocarcinoma to be described with anatomical precision was reported in 1997 by Sequens et al. [43]. Other cases have been reported since then [44,45,46,47,48,49,50,51,52,53,54]. The main published cohorts are summarised in Table 1.

The presence of dysplasia or cancer in the proctocolectomy specimen suggests that the patient is at risk of developing ATZ cancer [26,34,55,56]. Pedersen et al. described a case of ATZ adenocarcinoma following pouch excision without abdominoperineal excision [57]. RPC was originally carried out for high-grade rectal dysplasia. In view of the risk, total abdominoperineal excision should be discussed for patients requiring pouch removal. The duration of UC has also been identified as a risk factor for ATZ cancer, such as colorectal cancer [58]. Finally, no link has been established between the risk of ATZ dysplasia/cancer and the type of anastomosis performed (hand-sewn with mucosectomy/stapled) [42].

**Table 1 cancers-14-00530-t001:** Main published studies estimating the risk of ATZ dysplasia and cancer in patients with ulcerative colitis, including clinical and endoscopic follow-up.

Author, Year, [Ref.]	Design	Location of the Study	Pre-Op Colonic Dysplasia/Cancer	Number of Patients	Mean Follow-Up (Months)	Number of Cases	Identified Risk Factors
Tsunoda et al., 1990 [26]	Retrospective	England	10.2% of dysplasia, 6.8% of cancer	118	NA	3 dysplasia	Duration of the disease > 10 years Pre-op cancer
Schmitt et al., 1992 [59]	Prospective	USA	NA	50	8.6	0 dysplasia, 0 cancer	none
Ziv et al., 1994 [55]	Retrospective	USA	9.4% high-grade dysplasia, 4.3% cancer	254	28	8 low-grade dysplasia	Pre-op dysplasia or cancer
Haray et al., 1996 [60]	Retrospective	USA	1 patient with dysplasia	109	31	0 dysplasia, 0 cancer	none
Sarigol et al., 1999 [61]	Prospective	USA	6.6% dysplasia	76	60	0 dysplasia	none
O’Riordain et al., 2000 [56]	Retrospective	USA	10.5% dysplasia, 4.3% cancer	210	77	6 low-grade dysplasia, 1 high-grade dysplasia	Pre-op dysplasia or cancer
Remzi et al., 2003 [34]	Prospective	USA	14.6% dysplasia, 3.4% cancer	178	130	6 low-grade dysplasia, 2 high-grade dysplasia	Pre-op dysplasia or cancer
Kayaalp et al., 2003 [62]	Retrospective	Turkey	9.1% dysplasia	44	42	1 dysplasia	none
Kariv et al., 2010 [63]	Retrospective	USA	13.7% dysplasia, 1.8% cancer	3203	99	16 dysplasia, 10 cancers	Pre-op dysplasia or cancer
Mathis et al., 2011 [64]	Retrospective	USA	47% dysplasia, 9% cancer	100	71	1 adenocarcinoma	none
Zhu et al., 2013 [65]	Retrospective	USA	13.8% dysplasia, 4.3% cancer	123	89	1 indeterminate for dysplasia	none
Silva-Velazco et al., 2014 [66]	Retrospective	USA	10.5% dysplasia, 2.5% cancer	285	125	6 low-grade dysplasia, 3 high-grade dysplasia	Pre-op dysplasia or cancer
Block et al., 2015 [67]	Prospective	Sweden	46% low-grade dysplasia, 34% high-grade dysplasia, 20% cancer	56	216	4 indefinite for dysplasia	none
Lightner et al., 2020 [68]	Retrospective	USA	NA	3672	NA	7 low-grade dysplasia, 4 cancers	none

NA: not available; Pre-op: pre-operative.

Ravitch gave the first description of adenocarcinoma of the pouch, per se, following RPC for UC in 1984 [69]. Thirteen case reports have been published since then [70,71,72,73,74,75,76,77,78,79,80,81,82]. The main cohorts reported on are summarised in Table 2. The largest study estimated the risk of developing pouch cancer to be 0.2%, 0.4%, 0.8%, 2.4% and 3.4% 5, 10, 15, 20 and 25 years post-RPC, respectively, [63]. The cumulative incidence for pouch dysplasia at 5, 10, 15, 20 and 25 years was 0.8%, 1.3%, 1.5%, 2.2% and 3.2%, respectively.

The only risk factor identified for pouch cancer in this study was pre-operative colorectal cancer (adjusted hazard ratio of 13.43; *p* < 0.001) and dysplasia (adjusted hazard ratio of 3.62; *p* = 0.002). A link with primary sclerosing cholangitis has been put forward as a risk factor for pouch dysplasia and cancer [83]. Similar to ATZ neoplasia, the duration of UC has also been described as a risk factor [84,85]. Some studies suggested that chronic pouchitis might be a risk factor for dysplasia onset [85,86]. However, the largest available cohorts showed no link between pouchitis and dysplasia (or cancer) of the pouch [29,63]. Finally, pre-operative backwash ileitis has been put forward as a risk factor for pouch dysplasia [72]. However, other studies found no link with pouch dysplasia [87].

**Table 2 cancers-14-00530-t002:** Main published studies estimating the risk of pouch dysplasia and cancer in patients with ulcerative colitis, including adequate clinical and endoscopic follow-up.

Author, Year, [Ref.]	Design	Location of the Study	Pre-Op Colonic Dysplasia/Cancer	Number of Patients	Mean Follow-Up (Months)	Number of Cases	Identified Risk Factors
Emblem et al., 1988 [88]	Prospective	Norway	10.5% low-grade dysplasia	19	>36	0 dysplasia	none
Setti Carraro et al., 1994 [89]	Retrospective	England	23.3% dysplasia	60	97	0 dysplasia	none
Gullberg et al., 1997 [86]	Prospective	Sweden	20% low-grade dysplasia, 12% high-grade dysplasia	25	54	4 low-grade dysplasia, 1 high-grade dysplasia	Severe villous atrophy
Sarigol et al., 1999 [61]	Prospective	USA	6.6% dysplasia	76	60	0 dysplasia	none
Ettorre et al., 2000 [90]	Prospective	France/Italy	NA	21	85	0 dysplasia	none
Tiainen et al., 2001 [91]	Retrospective	Finland	NA	36	118	0 dysplasia	none
Heuschen et al., 2001 [92]	Retrospective	Germany	7.2% dysplasia, 9.5% cancer	308	48	1 cancer	none
Thompson-Fawcett et al., 2001 [93]	Prospective	Canada	NA for dysplasia, 10.4% cancer	106	>91	1 low-grade dysplasia	none
Hultén et al., 2002 [94]	Retrospective	Sweden	NA	40	360	3 low-grade dysplasia	none
Herline et al., 2003 [95]	Retrospective	USA	NA	160	101	1 low-grade dysplasia	none
Ståhlberg et al., 2003 [83]	Prospective	Sweden	43.8% dysplasia	32	144	5 low-grade dysplasia, 1 high grade dysplasia	primary sclerosing cholangitis
Börjesson et al., 2004 [96]	Prospective	Sweden	0 cancer	45	192	2 low-grade dysplasia	none
Nilubol et al., 2007 [97]	Prospective	USA	15.9% dysplasia, 5.8% cancer	118	65	1 indeterminate for dysplasia	none
Zmora et al., 2009 [98]	Retrospective	Israel	8.6% dysplasia, 7.6% cancer	185	97	1 cancer	none
Kariv et al., 2010 [63]	Retrospective	USA	13.7% dysplasia, 1.8% cancer	3203	99	8 dysplasia, 3 cancers	Pre-op dysplasia or cancer
Al-Sukhni et al., 2010 [99]	Retrospective	Canada	72.8% dysplasia, 27.2% cancer	81	76	1 dysplasia, 1 cancer	none
Hernández et al., 2010 [100]	Prospective	Puerto Rico	26% dysplasia	38	12	1 low-grade dysplasia	none
Burdyński et al., 2011 [101]	Retrospective	Poland	NA	87	139	2 low-grade dysplasia, 1 high-grade dysplasia	none
Shuno et al., 2011 [102]	Retrospective	Japan	NA	68	64	1 low-grade dysplasia, 1 high-grade dysplasia	none
O’Riordain et al., 2012 [58]	Retrospective	Canada	NA	2010	161	0.0015% of cancer	none
Kuiper et al., 2012 [103]	Prospective	The Netherlands	27.3% low-grade dysplasia, 18.2% high-grade dysplasia, 25% cancer	44	103	2 low-grade dysplasia	none
Derikx et al., 2014 [29]	Retrospective	The Netherlands	9.4% dysplasia, 4.2% cancer	1200	100	8 low-grade dysplasia, 1 high-grade dysplasia, 16 cancers	Pre-op dysplasia or cancer
Imam et al., 2014 [104]	Retrospective	USA	41.5% low-grade dysplasia, 6.2% high-grade dysplasia, 10.8% cancer	65	72	1 low-grade dysplasia, 1 high-grade dysplasia, 1 cancer	primary sclerosing cholangitis
Bobkiewicz et al., 2015 [85]	Retrospective	Poland	20.3% low-grade dysplasia, 9.1% high-grade dysplasia, 1.8% cancer	276	118	5 low-grade dysplasia, 3 high-grade dysplasia, 1 cancer	Pre-op dysplasia or cancer, duration of UC, duration of follow-up, pouchitis
Block et al., 2015 [67]	Prospective	Sweden	46% low-grade dysplasia, 34% high-grade dysplasia, 20% cancer	56	216	20 indefinite for dysplasia, 1 low-grade dysplasia	None
Ishii et al., 2016 [105]	Retrospective	Japan	27% dysplasia or cancer	90	120	1 low-grade dysplasia, 1 cancer	None
Mark-Christensen et al., 2018 [106]	Retrospective	Denmark	1.1% cancer	1723	155	2 cancers	None
Lightnert et al., 2020 [68]	Retrospective	USA	NA	3672	NA	2 cancers	none

NA: not available; Pre-op: pre-operative.

More anecdotally, other pouch and ATZ malignancies have been reported such as squamous cell carcinoma [63,66,69,106,107,108,109,110], pouch lymphoma [29,63,99,111,112,113,114,115,116,117], carcinoid tumour of the pouch [118,119] and malignant melanoma [120]. Moreover, the Danish nationwide cohort found a higher risk of non-melanoma skin and hepatobiliary cancers in patients undergoing IPAA for UC than in the gender- and age-matched comparison cohort of patients from a national civil database [106]. These results might be explained by the exposition to thiopurines and the co-existence of primitive sclerosing cholangitis, respectively. Similarly, an increased risk of renal cell cancer was highlighted in another study [121]. The main references in this topic could be as follows: [29,34,55,63,68,72,103].

### 3.2. Underlying Mechanisms

The risk of dysplasia and cancer of the ATZ and rectal cuff have been postulated more frequently in the case of stapled IPAA. Yet, stapled anastomosis does not seem to increase the risk of rectal cuff and ATZ cancers [84]. Given the better functional outcomes, stapled anastomosis is the preferred option according to the European guidelines [5]. The latter also state that the maximum length of anorectal mucosa left between the dentate line and anastomosis should not exceed 2 cm in order to lower the risk of cuffitis, dysplasia and cancer. Moreover, despite rectal mucosectomy, remaining islets of rectal mucosa can be found in up to 20% of patients after RPC with hand-sewn anastomosis for UC [122,123]. These data partly explain the rate of ATZ dysplasia and cancer observed in the case of hand-sewn anastomosis [58].

Furthermore, the presence of ATZ dysplasia and adenocarcinoma at the time of RPC has also been reported [124,125,126]. Sagayama et al. reported an incidence of preoperative dysplasia of ATZ at 4.4% [126]. This illustrates that, in some cases, the dysplasia might precede the IPAA. The third neoplastic pathway described is through chronic inflammation of the ATZ, which can be detected in up to four out of five patients after RPC for UC [127,128]. Similar to colitis-related colorectal cancer, the risk of ATZ dysplasia and cancer might increase in the context of chronic inflammation.

Veress et al. pointed out that some patients undergo mucosal adaptation with chronic villous atrophy after RPC (described as “type C” mucosal adaptation) [129]. This atrophy seems to be due to chronic inflammation [92,130,131]. Histologically, this adaptation (also called “colonic metaplasia”) is characterised by villous atrophy, crypt hyperplasia, neutrophilic and eosinophilic inflammation and increased Paneth and Goblet cell counts [132,133]. The number of patients with this type of mucosal adaptations increases the longer the follow-up period [134,135]. Changes in expression mucins have also been reported with an increase in colonic sulphomucins associated with the degree of villous atrophy and chronic inflammation [136,137,138].

This severe villous atrophy with chronic inflammation is assumed to highlight the risk of dysplasia and DNA aneuploidy [86,139]. Primary sclerosing cholangitis has also been linked to chronic villous atrophy [83]. This might partially account for the assumed higher risk of dysplasia and cancer in this subgroup of patients. However, atrophic pouch metaplasia is not always followed by dysplasia onset [96].

Microbiota are involved in the colonic metaplasia of the ileal reservoir [140,141,142]. Strict anaerobic bacteria predominate in the pouch, whereas facultative species predominate in UC patients with terminal ileostomy [143,144]. This change does not appear to occur in the case of FAP [143,145]. In the study by Kuisma et al., a higher overall faecal anaerobic bacterial count was associated with chronic villous atrophy and colonic metaplasia [146]. To highlight the role of the microbiota in the development of pouch dysplasia, changes in the mucosal pattern of ileal pouches, characterised by villous atrophy, have been reported to start as early as 6 days to 6 weeks after ileostomy suppression [147,148]. Das et al. studied the mucosal morphology of the pouch of patients with indefinite diversion (without pouch excision) [149]. None of the 20 patients developed dysplasia or pouch cancer (mean follow-up of 3.6 years after ileal diversion).

Sulphate-reducing bacteria are associated with greater sulphomucin expression in pouches [143]. Sulphate-reducing bacteria may interfere with goblet cell differentiation and, as a result of hydrogen sulphide production, may trigger epithelial apoptosis [143,150]. The latter phenomenon is mediated by inhibition of butyrate oxidation by hydrogen sulphide which impairs its use by epithelial cells [151]. Compared to non-colectomised patients, the concentration of short-chain fatty acids seems similar in pouches, albeit with an increased acetate ratio [144,152].

Depleted levels of secondary bile acids (lithocholic acid and deoxycholic acid) have been found in the pouch of UC patients compared to FAP patients [153]. Sinha et al. have proven that secondary bile acids have the capacity to mitigate inflammation in murine colitis models [153]. Finally, increased TLR4 expression (a bacterial antigen receptor) has been highlighted in endoscopically normal pouches compared to normal ileum [154]. This over-expression might be one of the pathways from dysbiosis to luminal inflammation.

The underlying pathology might also be involved in the risk of pouch inflammation, dysplasia and cancer. Indeed, compared to FAP patients, it has been proven that pro-inflammatory cytokines (such as TNFα) and pro-apoptotic proteins are up-regulated in endoscopically normal pouches of UC patients [155,156]. Modulation of autophagy markers in the ileal pouch of UC patients has also been described (decreased Beclin-1 protein levels) [157]. This might explain the higher incidence of pouch inflammation observed in UC patients.

Genetic aberrations are assumed to occur during the metaplasia-dysplasia-cancer sequence of the pouch, similar to colorectal carcinogenesis. The only modification identified by Gullberg et al. was the loss of heterozygosity at chromosome 5q14-22 [158]. Controversy surrounds the role of p53. Coull et al. found no correlation between ATZ dysplasia and overexpression of p53 (affecting one in two patients) [159]. On the contrary, other studies have highlighted a link between over-expressed p53 and aneuploidy, dysplasia and pouch cancer [160,161]. Finally, alterations in miRNA expression (mostly up-regulated) have been described in the pouches of UC patients [162,163].

The main references on all these underlying mechanisms could be as follows: [86,122,123,129,137,143,146].

### 3.3. Endoscopic Monitoring

Endoscopically, dysplasia and cancer of the pouch or ATZ can present as polypoid or non-polypoid lesions (including flat and ulcerated lesions) [164]. Only low-level evidence studies have been conducted as part of the endoscopic monitoring programme given the low prevalence of such complications. Hence, pouch monitoring remains controversial [165]. However, international guidelines based on risk stratification are available.

The British Society of Gastroenterology (BSG) recommends annual pouchoscopy for high-risk asymptomatic patients, defined by the presence of at least one of the following criteria: type C mucosal changes, primary sclerosing cholangitis or RPC for dysplasia or cancer [166]. The European Crohn’s and Colitis Organisation (ECCO) also recommends annual pouchoscopy for high-risk patients (same criteria and unremitting pouchitis) [167,168]. Both societies recommend pouchoscopy every 5 years for low-risk patients (Figure 2). According to the American Society of Gastrointestinal Endoscopy (ASGE), annual monitoring is mandatory in the case of RPC for dysplasia or cancer, and may be considered for patients with type C mucosal changes, primary sclerosing cholangitis and refractory pouchitis [169]. Pouchoscopy is also indicated if the following symptoms develop, namely diarrhea, haematochezia, abdominal pain, iron deficiency or anemia, etc. Despite those recommendations, an American survey reported by Gu et al. found heterogeneous practices concerning the endoscopy interval [170]. A European retrospective cohort study found similar results with one-third of the cohort who had never undergone pouchoscopy during follow-up (median duration of 10.5 years) [171].

More recently, consensus guidelines on the diagnosis and classifications of ileal pouch disorders have been published by the International Ileal Pouch Consortium [172]. Experts recommend a surveillance pouchoscopy programme depending on individual risk. Annual pouchoscopy is advocated in the case of pre-colectomy diagnosis of colitis-related dysplasia or cancer. Pouchoscopy every 1–3 years is recommended for patients with associated primary sclerosing cholangitis, chronic pouchitis (or cuffitis), Crohn’s disease of the pouch, persistent ulcerative colitis (≥8 years) and in the event of a family history of colorectal cancer. If none of the afore-mentioned risk factors is present, pouchoscopy can be performed every 3 years.

The therapeutic strategy is outlined in Figure 3. Although most of pouch polyps are inflammatory, polypectomy must be offered for pouch polyps exceeding 1 cm, for polyps localised on the ATZ/rectal cuff and symptomatic inflammatory polyps [173,174]. Repeated biopsies within 6 months might be advisable in case of low-grade dysplasia. Indeed, regression of low-grade dysplasia to normal mucosa has been reported in serial biopsies [30,34,55]. Regression has even been documented in high-grade dysplasia patients in some studies [29]. Such findings may be explained by sampling errors in an area where it is difficult to perform biopsies [175]. Therefore, multiple and repeated biopsies are mandatory on endoscopic examination of the pouch. Multiple biopsies (at least 3–4) must be taken from each of the following regions: the ATZ, pouch and afferent ileal limb [176,177]. The biopsies collected from each region must be analysed separately.

Moreover, Wu et al. suggested that a family history of colorectal cancer is associated with an increased risk of low-grade dysplasia progression [30]. Hence, such patients require increased vigilance. Finally, endoscopic mucosectomy is the preferred option for high-grade dysplasia and for recurrent (or persistently positive) low-grade dysplasia biopsies. The recent international consensus stated that patients with a history of dysplasia of the ATZ/rectal cuff (or pouch) must undergo close monitoring with early pouchoscopy 3–6 months after endoscopic treatment, and annually thereafter [172]. Complete surgical mucosectomy with pouch advancement (or pouch resection with terminal ileostomy) is indicated if endoscopic treatment fails or proves impossible.

The use of chromoendoscopy with targeted biopsies has been assessed in only one study which failed to highlight any benefits with this technique [103]. The use of high-magnification chromoscopic endoscopy has been recommended, but it has never been put to widespread use [178]. One way of improving the detection of cancer and dysplasia during endoscopy would be to develop innovative tools based on artificial intelligence [179].

The main references on endoscopic screening could be as follows: [166,167,169,172,176].

## 4. Adenomas, Dysplasia and Cancer of the ATZ and Ileal Pouch in Patients with FAP

### 4.1. Epidemiology and Risk Factors

The cumulative risk of developing ATZ adenoma has been estimated by Van Duijvendijk et al. as 8 and 18% at 3.5 and 7 years post-surgery, respectively, [180]. Tsunoda et al. had previously reported on a retrospective cohort of patients who developed ATZ dysplasia (high-grade for 21.4% of patients) [26]. Hoehner and Metcalf provided the first description of ATZ adenocarcinoma following RPC for FAP in 1994 [181]. Since then, seven case reports have been published [182,183,184,185,186,187,188]. The main cohorts highlighting the risk of dysplasia and cancer of the ATZ after restorative proctocolectomy for FAP are summarised in Table 3. Due to the low incidence of dysplasia and adenocarcinoma of the ATZ, no risk factors have been identified to date. However, stapled anastomosis appears to be linked to a higher risk of ATZ adenomas [18,23,180,189]. Severe colic disease (>1000 polyps) was identified as a risk factor in ATZ adenomas in one study [23].

Numerous cases of pouch adenomas following RPC for FAP have been reported [18,23,190,192,196,197,198,199,200,201,202,203,204,205,206]. This risk increases over time and prevalence has been estimated as 7%, 35% and 75% at 5, 10 and 15 years post-surgery, respectively, [207]. The presence of gastric adenoma, male gender and ≤ 18 years old at the time of surgery have been reported as risk factors for pouch adenomas by Ganschow et al. [208]. Goldstein et al. reported a higher incidence of pouch adenomas in the case of related duodenal adenomas [205]. On the contrary, Wu et al. found no correlation between the severity of duodenal disease (according to Spigelman’s classification system) and pouch adenomas [190]. Severe colic disease (>1000 polyps) has been identified by Tonelli et al. as a risk factor for pouch adenomas.

High-grade dysplasia and adenocarcinomas of the pouch have been documented in case reports [209,210,211,212,213,214,215,216,217]. The main cohorts highlighting the risk of dysplasia and cancer of the pouch after restorative proctocolectomy for FAP are summarised in Table 4. In terms of risk factors, no correlation has been established between pouchitis and the onset of dysplasia [193]. Data regarding the overall risk of pouch dysplasia and cancer post-IPAA and the impact of the anastomosis technique are sparse. The meta-analysis conducted by Lovegrove reported a lower rate of dysplasia with hand-sewn anastomosis (7.2% vs. 18.5% with double-stapled anastomosis). However, this result was not significant (*p* = 0.08) [218].

The main references on epidemiology and risk factor could be as follows: [23,180,194,196,207].

### 4.2. Underlying Mechanisms

The presence of residual rectal mucosa is one of the main reasons behind the onset of pouch adenomas in the case of stapled anastomosis [223]. On the other hand, remaining islets of rectal mucosa have been described in patients with hand-sewn anastomosis, despite mucosectomy [123].

Despite removal of the entire rectal mucosa, colonic metaplasia of the ileal mucosa was still observed, as in UC [224,225]. An increased rate of asymmetrical fission of the pouch crypts has been put forward to explain this hyperplasia [226]. Some authors assume that colonic metaplasia is due to faecal stasis in the pouch, which alters the luminal content and, consequently, adapts the epithelium [22,227]. Indeed, the number of post-IPAA faecal bacteria is 10 times higher than that reported following terminal ileostomy [228]. Functional changes have also been observed such as a change from mucins to colonic sulphomucins and the increased metabolism of primary conjugated bile acids [228,229,230]. To substantiate the theory of colonic metaplasia, cases of terminal ileostomy adenocarcinoma, in which the mucosa adjacent to the tumour presented colonic mucosa characteristics, have also been reported [231,232].

Friedrich et al. demonstrated that Glutathione S-transferase activity, which has a protective role in carcinogenesis, is lower in the pouch than in the proximal ileum [233]. Paiva et al. found fewer autophagy markers (ATG5 and MAP1LC3A) in the ileal pouch mucosa of FAP [157]. The heightened risk of adenoma is also due to the increased proliferation rate of epithelial cells in the pouch [234].

A molecular study by Will et al. provides data on the APC mutation spectrum in pouch adenomas [235]. In FAP, there is a correlation between germline and somatic APC mutation. In their study, Will et al. proved that pouch adenomas are genetically closer to colorectal adenomas. Recently, Kariv et al. described a link between pouch and cuff adenoma and the type and location of APC mutation [206]. The risk of adenoma was increased in the event of indel/deletion mutation and a higher number of adenomas per patient was linked to exon 15 mutation. Conversely, Groves et al. found no link between genotypic characteristics and pouch adenomas [200].

Figure 4 shows the main risk factors, frequency and putative underlying mechanisms of high-grade dysplasia and cancer in the ATZ and ileal pouch following RPC for UC and FAP. The main references on all these underlying mechanisms could be as follows: [123,142,206,217]

### 4.3. Endoscopic Monitoring

This high incidence of adenomas and reported cases of cancer of the pouch emphasize the need for endoscopic monitoring. The European Society of Gastrointestinal Endoscopy (ESGE) published guidelines in 2019 [236]. It advocates endoscopic pouch monitoring every 1–2 years in the case of FAP (Figure 5). The ESGE recommends removing pouch polyps > 5 mm and all ATZ and rectal cuff polyps. Cold snare polypectomy is the technique mostly used to remove pouch polyps [237]. All of these recommendations are sound but rely on a poor evidence base.

Some authors even advocate initiating thorough endoscopy monitoring 6 months and 1 year after surgery with 2 yearly follow-ups thereafter [238,239]. The use of indigo carmine chromoendoscopy promotes the detection of small adenomas (<5 mm) [219,240]. Retroflexion in the pouch has been seen to increase the adenoma detection rate in the cuff/ATZ without any specific complications [241]. Yet, retroflexion might be painful for patients not under general anesthesia; therefore, its use must be adapted to patients’ tolerance. Concerning bowel preparation, as for UC, a single sodium phosphate enema is usually sufficient to allow a complete examination of the pouch.

The major limiting factor in endoscopic monitoring is poor patient compliance. Douma et al. estimated that 8% of pouch patients fail to comply with recommended monitoring procedures [242] mainly due to lack of symptoms and the unpleasant nature of endoscopy per se. The main references on this endoscopic screening could be as follows: [236,239,241,242].

## 5. Discussion

We have collated available data on dysplasia and cancer of the ATZ and pouch in the case of UC and FAP. The overall incidence of pouch adenomas was assessed in up to 75% of patients 15 years post-surgery [207]. The risk of dysplasia and cancer is due to a combination of genetic predisposition (germline *APC* mutation), intestinal epithelial changes due to faecal stasis and the presence of residual rectal mucosa [227]. Smith et al. estimated that 75% of pouch-related cancers post-RPC with IPAA in FAP patients are located in the ATZ [27]. Particular attention must be given to this area during patient follow-up (digital examination of the area, systematic biopsies and a retroflexion view of the pouch).

The incidence rate of pouch adenocarcinoma has been estimated at 0.35% 20 years after surgery in UC patients [28]. The underlying mechanisms in the UC context are partly based on the chronic inflammation-dysplasia-cancer pathway [243]. Chronic exposure of the ileal mucosa to dysbiotic microbiota led to colonic type phenotypic changes through modulation of the short fatty acid chains available, mucin secretion and bile acid metabolism. Despite the low incidence, the poor prognosis of pouch adenocarcinoma (evidenced by a 30% mortality rate 1 year after cohort diagnosis reported by Kariv et al.) confirms the major benefit of regular pouchoscopy [63].

Our review has several limitations. First of all, many of the studies reported have retrospective, small sample size cohorts with heterogeneous follow-up. The heterogeneity of the surgical techniques used in the studies creates further bias [18,124,181,191]. Much progress has been made since 1988 and the first prospective cohort reported by Emblem et al., which explains the lack of evidence regarding one of the anastomotic techniques used (stapled vs. hand-sewn) [88]. In most of the studies focusing on the risk of adenoma, dysplasia and cancer of the IPAA, it is impossible to differentiate between adenomas located in the anal transitional zone and those in the rectal cuff or pouch [205,219,244]. The prevalence of ATZ lesions may well be under-estimated.

Some studies assessing the risk of pouch adenomas in FAP patients do not specify the presence and extent of dysplasia within the adenomas [190,192,194,195]. Yet, in the case of FAP, almost all of the adenomas led to dysplasia (unlike inflammatory polyps for UC), thereby emphasising the importance of estimating the incidence of pouch adenomas, regardless of the degree of dysplasia. The difficulty of the histological examination is yet another limitation in terms of assessing dysplasia of the ATZ and pouch. Hultén et al. reported on a significant disagreement between 2 expert pathologists in reporting low-grade dysplasia for the same 40 IPAA biopsies [94]. The complexity of the pathological analysis stems from the frequent presence of acute inflammation and regenerating epithelium, which interfere with the diagnosis of dysplasia [245]. The frequent use of “indefinite for dysplasia” by pathologists to characterise pouch biopsies also illustrates that difficulty [246]. This interobserver variability in diagnosing dysplasia might partly account for the low incidence reported. Moreover, the quality of data given to the pathologist by the endoscopist (location of the biopsies, clinical characteristics, endoscopic description of the pouch) and the biopsy specimens per se are frequently poor [247]. The introduction of a standardised endoscopic reporting template has been instrumental in improving those outcomes [248].

Finally, some cohorts reported not one case of dysplasia or cancer of the ATZ (or cuff or pouch) despite long-term follow-up [61,249,250]. The relative rarity of the event and the absence of a standardised biopsy protocol might explain these findings. It was also difficult to identify risk factors to guide the clinician and provide personalised patient follow-up because of the low incidence.

## 6. Conclusions

ATZ and ileal pouch cancers following RPC for FAP (or UC) are seldom detected in patients with IPAA. However, in view of the high mortality rate associated with this complication and given that pouchoscopy is a straightforward monitoring option, we recommend regular endoscopic monitoring for those patients. The development of endoscopic therapeutic options (less invasive than surgery) to remove dysplasia before cancer onset, further corroborates the need for a monitoring programme.

Additional studies are required to improve our knowledge of the underlying mechanisms (colonic metaplasia, dysbiosis) that perpetuate the emergence of dysplasia and cancer. A greater understanding of the factors involved may lead to preventive treatments (probiotics, anti-inflammatory drugs, etc.). This approach will help practitioners to identify those patients requiring follow-up and to initiate personalised endoscopic monitoring programmes.

## Figures and Tables

**Figure 1 cancers-14-00530-f001:**
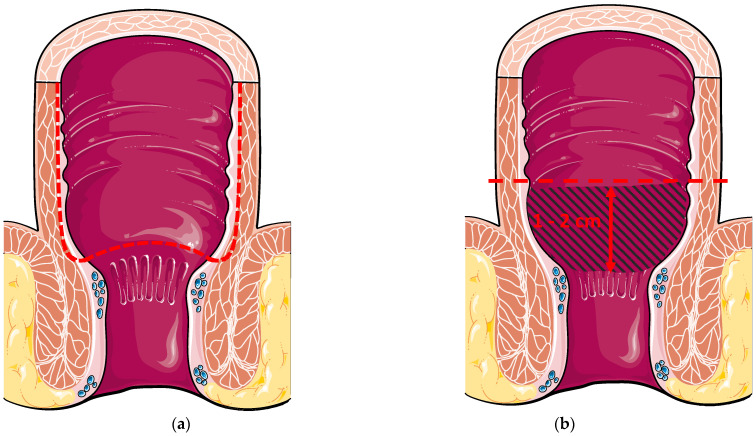
(**a**) Hand-sewn ileal pouch anal anastomosis with transanal mucosectomy. The dotted line represents the mucosectomy started above the dentate line; (**b**) Double-stapled ileal pouch anal anastomosis 1–2 cm above the dentate line. The blue hatched area represents the cuff rectal mucosa.

**Figure 2 cancers-14-00530-f002:**
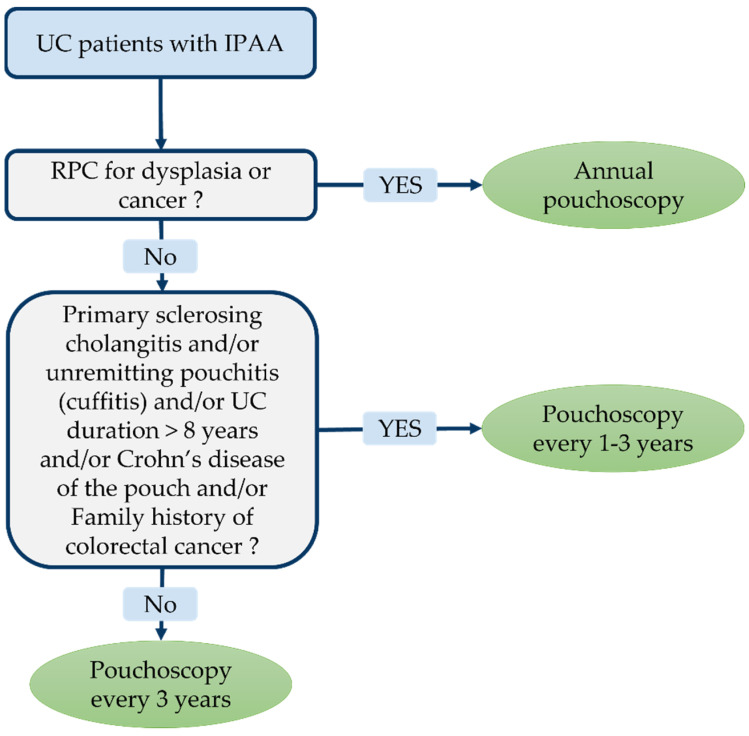
Endoscopic monitoring guidelines after restorative proctocolectomy (RPC) with ileal pouch-anal anastomosis (IPAA) for ulcerative colitis (UC) according the International Ileal Pouch Consortium [167,168].

**Figure 3 cancers-14-00530-f003:**
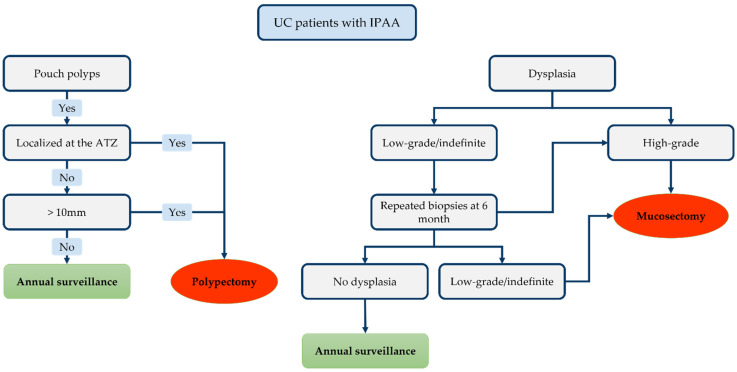
Proposed therapeutic algorithm for lesion detected on the ileal pouch-anal anastomosis (IPAA).

**Figure 4 cancers-14-00530-f004:**
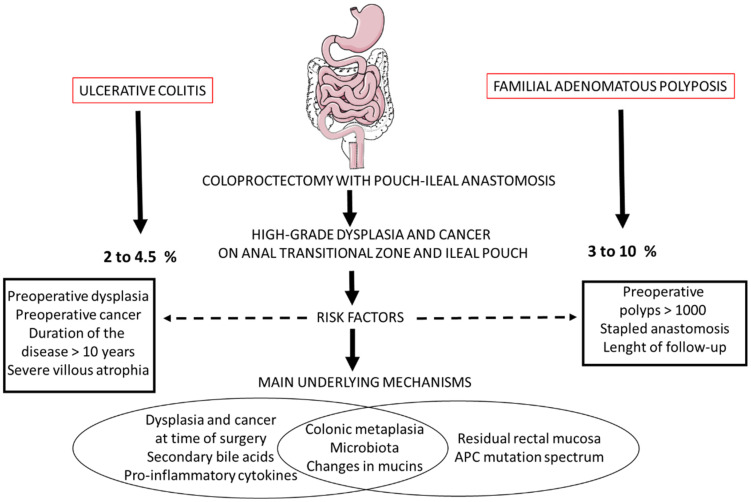
Main risk factors, frequency and putative underlying mechanisms of high-grade dysplasia and cancer in the anal transitional zone and ileal pouch following restorative coloproctectomy for ulcerative colitis and familial adenomatous polyposis.

**Figure 5 cancers-14-00530-f005:**
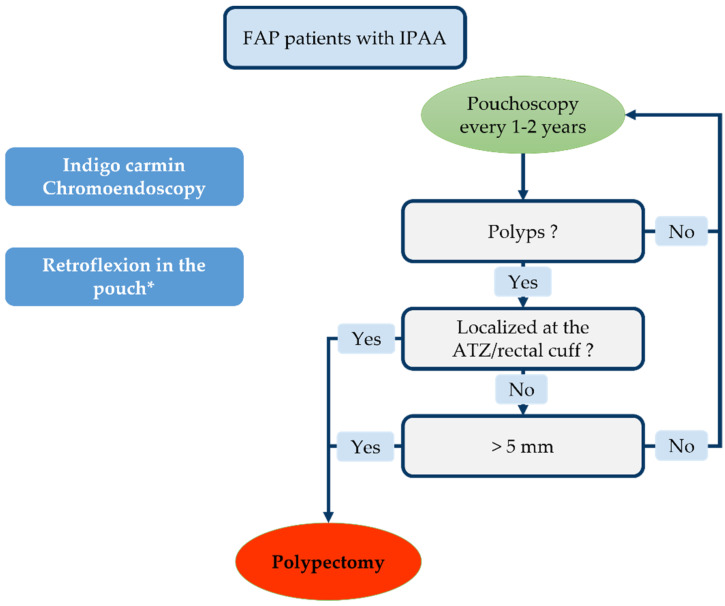
Endoscopic monitoring according to European Society of Gastrointestinal Endoscopy (ESGE) guidelines. * Depending on patient tolerance.

**Table 3 cancers-14-00530-t003:** Main published studies estimating the risk of adenomas, dysplasia and cancer of ATZ in patients with FAP including adequate clinical and endoscopic follow-up.

Author, Year, [Ref.]	Design	Location of the Study	Pre-Op Colonic Dysplasia/Cancer	Number of Patients	Median Follow-Up (Months)	Number of Cases	Identified Risk Factors
Tsunoda et al., 1990 [26]	Retrospective	England	100% of dysplasia, 28.6% of cancer	14	NA	12 dysplasia (3 high-grade)	none
Wu et al., 1998 [190]	Prospective	USA	NA	26	66	7 adenomas	none
Van Duijvendijk et al., 1999 [180]	Retrospective	The Netherlands	20.6% of cancer	97	66	13 adenomas (4 moderate dysplasia, 4 high-grade dysplasia)	Stapled anastomosis
Remzi et al., 2001 [23]	Retrospective	USA	NA	118	>42	27 adenomas, 1 cancer	Stapled anastomosis > 1000 polyps preoperatively
Ooi et al., 2003 [191]	Retrospective	USA	NA	148	NA	2 cancers	none
Von Roon et al., 2007 [189]	Retrospective	England	NA	140	123	52 adenomas (6 moderate dysplasia), 1 cancer	Stapled anastomosis
Booij et al., 2010 [192]	Retrospective	The Netherlands	77.8% of low-grade dysplasia, 11.1% of high-grade dysplasia, 11.1% of cancer	9	87	3 adenomas	none
Banasiewicz et al., 2011 [193]	Retrospective	Poland	NA	85	>24	19 low-grade dysplasia, 10 high-grade dysplasia, 1 cancer	none
Tonelli et al., 2012 [194]	Prospective	Italy	NA	69	133	3 adenomas	none
Wasmuth et al., 2013 [18]	Retrospective	Norway	NA	61	>164	18 adenomas, 1 cancer	Stapled anastomosis
Kennedy et al., 2014 [195]	Retrospective	USA	90% of low-grade dysplasia, 5% of high-grade dysplasia, 4.4% of cancer	95	91	9 adenomas	none
Lee et al., 2021 [196]	Retrospective	USA	17% of cancer	165	121	78 adenomas, 6 cancers	Stapled anastomosis

NA: not available.

**Table 4 cancers-14-00530-t004:** Main published studies estimating the risk of adenomas, dysplasia and cancer of the pouch in patients with FAP including adequate clinical and endoscopic follow-up.

Author, Year, [Ref.]	Design	Location of the Study	Pre-Op Colonic Dysplasia/Cancer	Number of Patients	Median Follow-Up (Months)	Number of Cases	Identified Risk Factors
Emblem et al., 1988 [88]	Prospective	Norway	23.1% of dysplasia, 23.1% of cancer	13	>36	10 adenomas (1 dysplasia)	none
Wu et al., 1998 [190]	Prospective	USA	NA	26	66	11 adenomas	none
Remzi et al., 2001 [23]	Retrospective	USA	NA	118	>42	23 adenomas	Stapled anastomosis
Groves et al., 2005 [200]	Prospective	England	NA	60	72	34 adenomas (23 low-grade dysplasia, 11 high-grade dysplasia)	Length of follow-up
Friederich et al., 2008 [219]	Retrospective	The Netherlands	NA	212	95	74 adenomas (25 high-grade dysplasia), 4 cancers	Stapled anastomosis
Campos et al., 2009 [220]	Retrospective	Brazil	60.2% of cancer	26	28	3 adenomas, 2 cancers	none
Booij et al., 2010 [192]	Retrospective	The Netherlands	77.8% of low-grade dysplasia, 11.1% of high-grade dysplasia, 11.1% of cancer	9	87	2 adenomas	none
Banasiewicz et al., 2011 [193]	Retrospective	Poland	NA	165	>24	13 low-grade dysplasia, 8 high-grade dysplasia, 5 cancers	none
Burdyński et al., 2011 [101]	Retrospective	Poland	NA	51	139	10 low-grade dysplasia, 5 high-grade dysplasia, 2 cancers	none
Tonelli et al., 2012 [194]	Prospective	Italy	NA	69	133	25 adenomas, 2 cancers	>50 years old at surgery >1000 polyps preoperatively
Wasmuth et al., 2013 [18]	Retrospective	Norway	NA	61	>164	14 adenomas	none
Boostrom et al., 2013 [221]	Retrospective	USA	NA	117	125	30 low-grade dysplasia, 1 cancer	None
Pommaret et al., 2013 [222]	Retrospective	France	NA	118	180	57 adenomas (7 high-grade dysplasia)	Duration of follow-up, advanced duodenal adenomas
Goldstein et al., 2015 [205]	Retrospective	Israel	NA	59	140	15 adenomas	Duodenal adenomas
Kariv et al., 2019 [206]	Retrospective	Israel	8.9% of high-grade dysplasia, 8.9% of cancer	45	>121	12 adenomas	Indel/deletion mutation of APC
Lee et al., 2021 [196]	Retrospective	USA	17% of cancer	165	121	47 adenomas	None

NA: not available.

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
