# Peer review of "Incidence and Risk Factors of Cancer in the Anal Transitional Zone and Ileal Pouch following Surgery for Ulcerative Colitis and Familial Adenomatous Polyposis"

_cancers, 2022, doi:10.3390/cancers14030530_

Round 1
Reviewer 1 Report
Title: Incidence and Risk Factors of Cancer in the Anal Transitional Zone and Ileal Pouch following Surgery for Ulcerative Colitis and Familial Adenomatous Polyposis
Thank you for the opportunity to review this manuscript. It proposes some interesting data and figures that are very usefull in daily practice. I really enjoyed reading it.
The authors offer an extensive review of the literature about incidence and risk factors of cancer after IPAA for FAP or UC. The bibliography and the references are exhaustive.
I only have a few remarks about spelling mistakes :
- Page 3, Line 6: replace IPPA by IPAA. Same correction in Figure 2.
- Figure 3: Under Pouch polyps, I would replace "No" by "Yes"
- Figure 4: Change Stappled into Stapled
Author Response
Thank you for the opportunity to review this manuscript. It proposes some interesting data and figures that are very useful in daily practice. I really enjoyed reading it.
The authors offer an extensive review of the literature about incidence and risk factors of cancer after IPAA for FAP or UC. The bibliography and the references are exhaustive.
I only have a few remarks about spelling mistakes:
- Page 3, Line 6: replace IPPA by IPAA. Same correction in Figure 2.
- Figure 3: Under Pouch polyps, I would replace "No" by "Yes"
- Figure 4: Change Stappled into Stapled
RESPONSE: We thank the reviewer for his remarks, all changes have been made accordingly (Page 3, Figures 2, 3, 4) (using “Track Changes” function of MS Word).
Reviewer 2 Report
Guillaume Le Cosquer et al. propose an interesting review about incidence and risk factors of cancer in the anal transitional zone and ileal pouch following surgery for UC and FAP. This review, very well written, can help gastroenterologists to manage these patients. This is a rare concern and literature is relatively poor on this topic. To join these two diseases (UC and FAP) in the same article also allow gastroenterologists to have an exhaustive review of this "anatomic situation". The presence of algorithms is another good aspect of this manuscript very - perhaps a bit too - complete: this allows the reader to have a visual summary of the work accomplished by the authors.
Some comments on this article before possible publication:
- The authors list in the introduction the short and long term complications of IPAA. It seems important to set aside the complications specific to the surgery following UC (pouchitis, development of Crohn's disease etc.), then to list the complications common to UC and FAP (incontinence, pouch dysplasia, etc.) . This would make reading easier.
- Figure 1 (although of good graphic quality) does not allow a clear visualization of the pouch. Making the J pouch appear clearly would make this figure clearer.
- Even if it’s a rare situation, some patients with Crohn’s disease have IPAA. Are they excluded of this study?
- The results of the review are well presented and the plan followed by the authors responds to an indisputable logic. Nevertheless, highlighting the main references (recent articles and / or with a high number of patients and / or prospective trials) at the beginning of each paragraph would allow the reader to have a clearer view of the "must-see" articles. For example (and this is an arbitrary choice), in section 3.1, references 33, 62 and 105 are particularly important: distinguishing these 3 articles between the others might be useful. It seems difficult, for example, to put on the same level of interest the reference 105 and the reference 120 as proposed at the end of section 3.1
- Section 3.2 is a bit long and would benefit from being more concise (especially if we compare to section 4.2). Even if an understanding of the underlying mechanisms is essential, this part can surely be a little reduced.
- Concerning endoscopic monitoring of patients with past UC (section 3.3): why the authors highlight ECCO’s recommendations in the figure 2? There are differences with the International Ileal Pouch Consortium, particularly for patients with no risk factors (every 3 years versus every 5 years). It is difficult to recommend a particular recommendation but the choice of the author for those of the ECCO must be justified.
- The issue of retroflexion in the pouch is a difficult one. Adherence to follow-up is difficult (and the authors emphasize this): retroflexion without anesthesia each year to patients followed for IPAA does not risk strengthening follow-up ... Should this point be mitigated in section 4.3 and figure 5? Should authors add that retroflexion is recommended particularly if the examination is performed under general anesthesia?
- It would also be interesting to have the position of the authors concerning the realization of the bowel preparation before the endoscopy of the pouch
Author Response
Guillaume Le Cosquer et al. propose an interesting review about incidence and risk factors of cancer in the anal transitional zone and ileal pouch following surgery for UC and FAP. This review, very well written, can help gastroenterologists to manage these patients. This is a rare concern and literature is relatively poor on this topic. To join these two diseases (UC and FAP) in the same article also allow gastroenterologists to have an exhaustive review of this "anatomic situation". The presence of algorithms is another good aspect of this manuscript very - perhaps a bit too - complete: this allows the reader to have a visual summary of the work accomplished by the authors.
Some comments on this article before possible publication:
- The authors list in the introduction the short- and long-term complications of IPAA. It seems important to set aside the complications specific to the surgery following UC (pouchitis, development of Crohn's disease etc.), then to list the complications common to UC and FAP (incontinence, pouch dysplasia, etc.). This would make reading easier.
RESPONSE 1: We thank the reviewer for his helpful remark, we acknowledge that postoperative complications must be split into two categories: disease related and common complications. Changes for clarification were made accordingly (page 2).
- Figure 1 (although of good graphic quality) does not allow a clear visualization of the pouch. Making the J pouch appear clearly would make this figure clearer.
RESPONSE 2: Thank you for this comment. We have decided to do not modify this figure as the exact purpose is the show clearly to the reader what difference exists between the two type of anastomosis, the mechanical one leaving clearly 1 to 2 cm of rectal mucosa above the anal transitional zone. In addition: ileal pouch is showed in figure 4.
- Even if it’s a rare situation, some patients with Crohn’s disease have IPAA. Are they excluded of this study?
RESPONSE 3: Given the rarity of the situation and the specific issues (high risk of fistula and perineal complications), we choose to exclude those patients from our study. This point was better explained (page 3) and the following reference added (i.e. new reference # 33):
Adamina, M.; Bonovas, S.; Raine, T.; Spinelli, A.; Warusavitarne, J.; Armuzzi, A.; Bachmann, O.; Bager, P.; Biancone, L.; Bokemeyer, B.; et al. ECCO Guidelines on Therapeutics in Crohn’s Disease: Surgical Treatment. Journal of Crohn’s and Colitis 2020, 14, 155–168, doi:10.1093/ecco-jcc/jjz187.
- The results of the review are well presented and the plan followed by the authors responds to an indisputable logic. Nevertheless, highlighting the main references (recent articles and / or with a high number of patients and / or prospective trials) at the beginning of each paragraph would allow the reader to have a clearer view of the "must-see" articles. For example (and this is an arbitrary choice), in section 3.1, references 33, 62 and 105 are particularly important: distinguishing these 3 articles between the others might be useful. It seems difficult, for example, to put on the same level of interest the reference 105 and the reference 120 as proposed at the end of section 3.1
RESPONSE 4: As suggested by the reviewer, we added a topic/sentence “Main references” in the last paragraph of each section.
- Section 3.2 is a bit long and would benefit from being more concise (especially if we compare to section 4.2). Even if an understanding of the underlying mechanisms is essential, this part can surely be a little reduced.
RESPONSE 5: we agree and the section 3.2 has been shortened (appeared using “Track Changes” function of MS Word)
- Concerning endoscopic monitoring of patients with past UC (section 3.3): why the authors highlight ECCO’s recommendations in the figure 2? There are differences with the International Ileal Pouch Consortium, particularly for patients with no risk factors (every 3 years versus every 5 years). It is difficult to recommend a particular recommendation but the choice of the author for those of the ECCO must be justified.
RESPONSE 6: We agree with the reviewer’s comment that it is difficult to recommend a particular recommendation over othersdue to lack of high quality of evidence-study (e.g. recommendations of follow up from the International Ileal Pouch consortium (IIPC) are grade B and based on 2.b evidence level studies).
We choose to highlight the ECCO’s recommendations because of their simple use with only two categories (annual vs every 5 years pouchoscopy). On the other hand, the IIPC recommendations have three categories: annual (in the case of pre-colectomy diagnosis of colitis-related dysplasia or cancer), every 1-3 years (primary sclerosing cholangitis, chronic pouchitis (or cuffitis), Crohn’s disease of the pouch, persistent ulcerative colitis (≥ 8 years) and family history of colorectal cancer) and every 3 years pouchoscopy (in the absence of risk factors). Yet, we acknowledge that IIPC recommendations are international (vs European), more recent (2021 vs 2015) and should therefore be emphasized. Figure 2 has been modified accordingly.
- The issue of retroflexion in the pouch is a difficult one. Adherence to follow-up is difficult (and the authors emphasize this): retroflexion without anesthesia each year to patients followed for IPAA does not risk strengthening follow-up ... Should this point be mitigated in section 4.3 and figure 5? Should authors add that retroflexion is recommended particularly if the examination is performed under general anesthesia?
RESPONSE 7: We thank the reviewer for his helpful remark. Indeed, we acknowledge that although retroflexion improves adenoma detection, it can be can painful and thereby reduce patients’ adherence to follow-up. Changes were made accordingly (page 14).
- It would also be interesting to have the position of the authors concerning the realization of the bowel preparation before the endoscopy of the pouch
RESPONSE 8: In our opinion, a single sodium phosphate enema is sufficient to allow a complete examination of the pouch. This point was added to the manuscript (page 14).
Reviewer 3 Report
Summary: This review by Le Cosquer and colleagues described the available data for dysplasia and cancer of the ATZ and pouch in patients with UC and FAP who underwent surgery (RPC/IPAA). I commend the authors for this review which I found to be comprehensive, thorough, and informative to read. The review is well written and contributes to the available literature on this subject. I was particularly interested in the post-IPAA endoscopic monitoring for this patient population which was well addressed by the authors. Below are suggestions for improving the manuscript:
Introduction: well written, no major comments.
Methods: How many independent authors carried out the PUBMED search?
Can the authors provide a Flow diagram for inclusion and exclusion of studies, including numbers?
Underlying mechanisms (UC):
In Page 7, the authors state “Furthermore, the presence of ATZ dysplasia and adenocarcinoma at the time of RPC has also been reported [123–125]. This illustrates that, in some cases, the dysplasia might precede the IPAA”. Did the authors find any available data on the incidence of ATZ dysplasia and adenocarcinoma (pre-surgery)? This would be nice to know in comparison to post-surgery incidence as described by this review.
Endoscopic monitoring (UC):
The authors described how often patients be offered endoscopic follow-up to screen for dysplasia and cancer of the ATZ or pouch based on current guidelines. Annual pouchoscopy for high risk and 5-yearly for low risk. Are there guidelines for stopping surveillance pouchoscopy?
Figure 2 and 5: ileal pouch-anal anastomosis (IPAA) or IPPA (typo?)
Discussion:
“Chronic exposition” should be rephrased (chronic exposure sounds better grammatically).
Overall comment: can the authors shorten the results section for readability? The review is rather too long to read. It would be nice if the authors can be more concise, particularly in sections 3 and 4 where the studies are reported.
Author Response
Summary: This review by Le Cosquer and colleagues described the available data for dysplasia and cancer of the ATZ and pouch in patients with UC and FAP who underwent surgery (RPC/IPAA). I commend the authors for this review which I found to be comprehensive, thorough, and informative to read. The review is well written and contributes to the available literature on this subject. I was particularly interested in the post-IPAA endoscopic monitoring for this patient population which was well addressed by the authors. Below are suggestions for improving the manuscript:
Introduction: well written, no major comments.
Methods: How many independent authors carried out the PUBMED search?
RESPONSE 1: Two authors (a gastroenterologist GLC and a digestive surgeon EB) conducted two independent literature searches in this systematic review, both using the same strategy. This has been was added (page 3).
Can the authors provide a Flow diagram for inclusion and exclusion of studies, including numbers?
RESPONSE 2: All articles from 1978 to December 2021 were included if they reported findings relating to dysplasia or cancer of the pouch and/or the ATZ and/or the rectal cuff (epidemiology, mechanisms, treatments, screening program) in context of UC or FAP. This point is now better explained on page 3.
Underlying mechanisms (UC):
In Page 7, the authors state “Furthermore, the presence of ATZ dysplasia and adenocarcinoma at the time of RPC has also been reported [123–125]. This illustrates that, in some cases, the dysplasia might precede the IPAA”. Did the authors find any available data on the incidence of ATZ dysplasia and adenocarcinoma (pre-surgery)? This would be nice to know in comparison to post-surgery incidence as described by this review.
RESPONSE 3: According to the largest and most recent study on this topic, the preoperative incidence of dysplasia of ATZ is around 4% (reference 126 within the manuscript). This point has been added in the manuscript.
Endoscopic monitoring (UC):
The authors described how often patients be offered endoscopic follow-up to screen for dysplasia and cancer of the ATZ or pouch based on current guidelines. Annual pouchoscopy for high risk and 5-yearly for low risk. Are there guidelines for stopping surveillance pouchoscopy?
RESPONSE 4: Neither ASGE, BSG and ECCO recommendations nor consensus guidelines from the International Ileal Pouch Consortium fixed this issue (references 166, 168, 169, 172 within the manuscript). As cases of adenocarcinoma have been diagnosed more than 20 years after surgery, we don’t recommend stopping pouch surveillance.
Figure 2 and 5: ileal pouch-anal anastomosis (IPAA) or IPPA (typo?)
RESPONSE 5: We thank the reviewer for his remark, changes have been made accordingly
Discussion:
“Chronic exposition” should be rephrased (chronic exposure sounds better grammatically).
RESPONSE 6: We thank the reviewer for his remark, change has been made accordingly.
Overall comment: can the authors shorten the results section for readability? The review is rather too long to read. It would be nice if the authors can be more concise, particularly in sections 3 and 4 where the studies are reported.
RESPONSE 7: sections 3 and 4 have been shortened (appeared using “Track Changes” function of MS Word).